# Experiences and challenges of people living with multiple long-term conditions in managing their care in primary care settings in Kerala, India: A qualitative study

Linju Joseph[1,2], Athira Krishnan[1], Thoniparambil Ravindranathanpillai Lekha[1], Neethu Sasidharan[1], Jissa Vinoda Thulaseedharan[1], Mathew Joseph Valamparampil[1], Sivadasanpillai Harikrishnan[1], Sheila Greenfield[2], Paramjit Gill[3], Justine Davies[2], Semira Manaseki-Holland[2‡], Panniyammakal Jeemon[1,2‡]*

1 Sree Chitra Tirunal Institute for Medical Sciences and Technology, Trivandrum, India, 2 Institute of Applied Health Research, College of Medical and Dental Sciences, University of Birmingham, Birmingham, United Kingdom, 3 Academic Unit of Primary Care (AUPC) Warwick Medical School, University of Warwick, Coventry, United Kingdom

‡ SMH and PJ are joint senior authors on this work.
* jeemon@sctimst.ac.in, p.jeemon@bham.ac.uk

## Abstract

### Background

Multimorbidity or multiple long-term conditions (MLTCs), the coexistence of two or more chronic conditions within an individual, presents a growing concern for healthcare systems and individuals' well-being. However, we know little about the experiences of those living with MLTCs in low- and middle-income countries (LMICs) such as India. We explore how people living with MLTCs describe their illness, their engagements with healthcare services, and challenges they face within primary care settings in Kerala, India.

### Methods

We designed a qualitative descriptive study and conducted in-depth, semi-structured interviews with 31 people (16 males and 15 females) from family health centres (FHCs) in Kerala. Interview data were recorded, transcribed, and thematic analysis using the Framework Method was undertaken.

### Findings

Two main themes and three sub-themes each were identified; (1) Illness impacts on life *(a) physical issues (b) psychological difficulties (c) challenges of self-management* and (2) Care-coordination maze *(a)fragmentation and poor continuity of care (b) medication management; an uphill battle and (c) primary care falling short*. All participants reported physical and psychological challenges associated with their MLTCs. Younger participants reported difficulties in their professional lives, while older participants found household activities challenging. Emotional struggles encompassed feelings of hopelessness and fear rooted in

**Data Availability Statement:** All relevant data (excerpts from the interview transcripts) are within the manuscript and its supporting information files.

**Funding:** A research grant from the Medical Research Council UK funded this work (MC_PC_MR/T037822/1). The funders had no role in study design, data collection and analysis, decision to publish, or preparation of the manuscript.

**Competing interests:** The authors have declared that no competing interests exist.

concerns about chronic illness and physical limitations. Older participants, adhering to Kerala's familial support norms, often found themselves emotionally distressed by the notion of burdening their children. Challenges in self-management, such as dietary restrictions, medication adherence, and physical activity engagement, were common. The study highlighted difficulties in coordinating care, primarily related to traveling to multiple healthcare facilities, and patients' perceptions of FHCs as fit for diabetes and hypertension management rather than their multiple conditions. Additionally, participants struggled to manage the task of remembering and consistently taking multiple medications, which was compounded by confusion and memory-related issues.

## Conclusion

This study offers an in-depth view of the experiences of individuals living with MLTCs from Kerala, India. It emphasizes the need for tailored and patient-centred approaches that enhance continuity and coordination of care to manage complex MLTCs in India and similar LMICs.

## Background

Multimorbidity or multiple long-term conditions (MLTCs), the co-existence of two or more chronic conditions within an individual, is a growing concern in healthcare, impacting individuals' well-being and healthcare systems worldwide [1]. MLTCs encompass various chronic conditions, including non-communicable diseases (NCDs), chronic infectious diseases, and mental health conditions [2]. In the context of the recent epidemiological transition, NCDs have emerged as the predominant contributors to mortality and disability, particularly in low- and middle-income countries (LMICs) [3, 4]. Notably, in LMIC settings such as India, NCDs tend to affect individuals at a younger age compared to high-income countries (HICs), leading to prolonged exposure to risk and, the co-occurrence of MLTCs [4]. In HICs, MLTCs are associated with higher rates of disability, reduced quality of life, and increased healthcare expenditures and utilization [5–7]. In LMICs, identified challenges from research studies include difficulties in accessing medicines, delays to care, and increased demand for healthcare services due to multiple referrals, leading to higher out-of-pocket expenditures [8–10].

In LMICs, one-fifth of the adult population is estimated to have a MLTC [11]. A cross-sectional study using the National Family Health Survey (NFHS)-4 reported a prevalence of 7.2% for MLTCs among adults aged 15–49 years in India and NFHS -5 showed that the prevalence of multimorbidity increased with advancing age [12, 13]. However, there are substantial variations in the reported prevalence of MLTCs in India, ranging from national reports of 7.2% [12] to 28% in Odisha [14] and 30% among older adults (age >60 years) in seven states [15]. In Kerala, there is a shortage of epidemiological data on MLTCs. Among the research studies available, one recent cross-sectional study conducted in a southern district of Kerala reported a community-level prevalence of 45% for MLTCs [16]. Another study revealed that two out of five participants seeking care at Family Health Centres (FHCs) in the Malappuram district of Kerala had MLTCs, and individuals with MLTCs reported a lower quality of life compared to those with no or a single chronic condition [17].

The Government of Kerala introduced localised government-sponsored pilots (*Amrutham Arogyam*) and the *Aardram Mission* in 2017 to strengthen primary health centres, especially services for hypertension and diabetes, through increased working time and improved staff

structure [18]. Further services such as dedicated NCD clinics, mental health clinics, and respiratory health clinics are also being implemented [19]. However, there remains a gap in the integrated management of MLTCs within primary care settings in Kerala or elsewhere in India.

To ensure services are improved commensurate with need requires understanding about the experience of living with MLTCs in India. Unfortunately, there is little knowledge in this area. Policies on primary care settings or NCDs in LMICs often fail to consider the patient's perspective, which is crucial for understanding their experiences of MLTCs and addressing broader social and economic factors that impact healthcare delivery, patient outcomes, and prognosis [20]. This study explores the experiences and perceptions of people living with MLTCs in Kerala. Specifically, it explores how people living with MLTCs describe their illness, their interactions with healthcare services, and their challenges in primary care settings in Kerala, India.

## Methods

### Study design and settings

This qualitative study was part of the broader research project titled "Systems thinking approach to developing an integrated and patient-centred intervention model for multimorbidity care in primary care settings in India."[21] An exploratory descriptive [22, 23] qualitative study was conducted among people living with MLTCs to explore their experiences, challenges, and care in primary care settings in Kerala, India.

Given the limited existing research on the subject, a qualitative descriptive approach was deemed appropriate. This methodology allows researchers to remain closely connected to the raw data, facilitating the emergence of insights into the phenomenon under investigation [23]. By adopting this approach, the study sought to provide an in-depth summary of the phenomenon through the analysis and interpretation of the meaning's individuals attribute to events and supported by reference to verbatim quotations from participants [24]. The overarching aim of the project was to enhance healthcare practice by developing an intervention model for managing patients with MLTCs.

Adopting a flexible design such as qualitative description enables data collection and analysis to be an iterative process by responding to participants' responses to questions and simultaneously adapting the analytical process as new insights emerge during the study [25]. Given the emphasis on providing a rich, descriptive account in qualitative descriptive design, thematic analysis using the Framework Method was chosen [26, 27]. The Framework Method is employed for its suitability for use in multidisciplinary research teams and flexibility for both inductive and deductive analysis. It allows for a systematic exploration of different views or experiences of participants in relation to each topic, which can then be compared and contrasted effectively. We have reported the qualitative study using SRQR (Standards for Reporting Qualitative Research) checklist.

The study was carried out in family health centres (FHCs or upgraded primary health care centres) located in the Northern (n = 3), Central (n = 3), and Southern (n = 2) districts of Kerala from 10th June 2022 to 10th December 2022. These areas were selected to encompass a diverse range of FHCs across different geographical regions in Kerala.

### Study participants, sampling, and recruitment

We used purposive sampling [28]. to identify individuals who met the inclusion criteria, which required that patients were adults (18 years and above), seeking care at FHCs, and had two or more chronic conditions. Patients considered too ill to participate by the nurses at the FHCs

were excluded. Eligible participants were recruited either in person at the FHCs or through telephone numbers provided by FHCs.

## Data collection

In-depth semi-structured interviews were conducted by a team of four researchers (LJ, LTR, AK, and NS), all of whom had prior experience in either public health or sociology and were trained in qualitative methodology. The researchers conducted the interviews face-to-face or over the phone, based on participant's preference. Interviews were conducted in Malayalam using a topic guide (S1 Box in S1 File). The topic guide included open-ended questions that encouraged participants to share their current health conditions, self-management practices, challenges related to their health and healthcare visits, communication with healthcare providers, their experiences, and opinions regarding the services available at the FHCs. The topic guide was prepared by consulting the research team and reviewing the literature on multimorbidity. Data collection and analysis were iterative. All the face-to-face interviews took place in private rooms within the FHCs to ensure confidentiality. Field notes were taken after each interview to record key issues and observations. All interviews were audio-recorded and varied in length, lasting between 15 and 90 minutes. In addition, two participants stated they had limited time; hence researchers ended the interviews based on participants' preference.

## Data analysis

Data analysis was conducted using the Framework method [27], which allowed for developing an analytical framework based on codes derived from the interview data and key areas of the topic guide. The analysis involved several stages: transcription, familiarization, coding, identifying an analytical framework, charting, mapping, and interpretation (See S1 Table in S1 File). Open coding or unrestricted coding facilitated excerpts from the interview data to be labelled inductively using qualitative data software Taguette [29, 30] Additionally, deductive codes from the topic guide were also included in the development of initial analytical framework. Two authors (LJ and AK) independently coded initial four interview data and discussed with another researcher (LTR) to understand the similarities and differences in coding and incorporate different views to develop an initial coding framework. Initial coding, analytical framework, charted data and interpretations were presented regularly to the team, throughout the analysis period.

The research team ensured not only code saturation but also a deep and rich understanding of the themes derived during analysis (meaning saturation) [31]. Regular meetings of the wider research team were held throughout data analysis to facilitate further exploration of participants' responses and agreement on recurring themes.

## Methodological rigour and the research team

We acknowledge that the use of the Framework Method may have led to the analysis and subsequent themes being influenced by the subjective interpretations of the research team. However, throughout the analytical process, researcher reflexivity and discussions occurred among the authors during data collection, analysis, and write-up to ensure rigor in the qualitative analysis conducted. Specifically, charting facilitated discussions among the diverse research team, and analyst triangulation was ensured, promoting objectivity between the researcher's position and the analysis. Charting made it easy to identify relevant quotes to illustrate themes from a range of participants and served as an audit trail from raw data to final themes [27]. Additionally, incorporating both experienced and novice qualitative researchers enriched the study by capturing diverse perspectives [32, 33]. The iterative approach to data collection and

analysis ensured thorough engagement with the data, enhancing the credibility and depth of insights gathered.

LJ was a female, early career researcher with specific interests in intervention development and patient experiences. LTR was a female sociologist, with research expertise and interests in healthcare. AK and NS were female research assistants with background in public health and nursing. PJ was a researcher with expertise in chronic disease epidemiology and health systems interventions. LJ, LTR, AK, NS and PJ are native of Kerala. SG is a professor of medical sociology with expertise in cross-cultural research from UK. The other team members (JVT, MJV, SH, PG, JD and SMH) had expertise in clinical and/health systems research.

## Ethics

We obtained Institutional Ethical Committee approval for the study from the Sree Chitra Tirunal Institute for Medical Science and Technology, Trivandrum (IEC/1543). Patients who met the inclusion criteria were provided with verbal and documented study information. The researchers obtained verbal and written informed consent from all participants and were informed that their involvement in the study would not change their clinical care. All data were anonymized before data analysis.

## Findings

We interviewed 31 participants with more than two long-term chronic conditions. The age of participants ranged from 41 to 82 years (Table 1). Of the 31 participants, 15 were female and 16 males. All study participants reported two or more conditions; 24 reported three or more chronic conditions. In our sample, the most common long-term conditions reported were diabetes (27 of 31 participants) and hypertension (27 of 31 participants). Further details can be found in S2 Table in S1 File.

Our study showed several barriers to seeking and managing care for participants with MLTCs in Kerala, India. The barriers experienced were organized into two themes: illness impacts on life and care-coordination maze. Two themes had further sub-themes, which are shown in Fig 1. Themes are supported with participant quotes, presented in clean verbatim style to improve readability. Additional illustrative quotes are given in S3 Table in S1 File.

## Illness impacts on life

Within our sample, people living with MLTCs conveyed experiencing both physical and psychological challenges related to their health. This was observed across participants of various age groups, with older and younger individuals reporting these dual issues and challenges to their self-management.

**Table 1. Demographic details of included participants.**

| Average age (SD) | 60.97 (10.4) |
| --- | --- |
| **Gender** | |
| Male | 16 (51.6%) |
| Female | 15 (48.3%) |
| **No: of chronic conditions** | |
| 2 | 7 (22.5%) |
| 3 | 9 (29%) |
| >3 | 15 (48.3%) |

**Fig 1. Participant perceptions of challenges for managing their multiple conditions.**

### Physical issues

The participants reflected on the difficulties they have faced in their daily lives after developing MLTCs. Most participants highlighted the physical limitations, such as limited range of motion and slowness in walking due to MLTCs and how they have impacted their ability to perform everyday tasks and responsibilities. Some participants also highlighted that they experienced disturbed sleep due to their symptoms or possible side effects of their multiple medicines.

> *"I cannot walk, feel palpitation, and have numbness in my hands. It is difficult for me to peel onions, clean and cut fish, and to sit [. . .]. Today, I am feeling much better. Or else it is tough." (P23, a 46-year-old female with six chronic conditions)*

Younger adult participants reported changing their work or giving up job entirely due to these limitations. In comparison, many older participants reported having limited abilities to perform household activities and facing difficulties in their daily lives. Some participants also highlighted the dependence on other family members to perform everyday tasks.

*"I don't work. I used to go to work, but now my family does not allow me to do work because of these diseases. I used to go for providing postnatal care and newborn care involving baby baths. So now I cannot lift babies, I am fragile so I quit my job." (P11, a 45-year- old female with four chronic conditions)*

*"I cannot lift heavy objects or engage in any physically exerting work. I used to do some agriculture-related activities around my house, which involved planting, cutting, and other related tasks. We also have some plantain plants near our house. Unfortunately, I am no longer able to perform these activities." (P9, a 62-year-old male with three chronic conditions)*

## Psychological difficulties

In our sample, people living with MLTCs were broadly reported having some psychological difficulties with MLTCs. Participants reported negative thoughts such as hopelessness and fear of being chronically ill due to the co-existence of multiple conditions, which led to emotional issues. Both younger and older participants described emotional difficulties due to their physical limitations, loss of work, and financial constraints caused by long-term treatment. Additionally, older participants reported emotional problems due to their increased dependence on their children. In this study, having multiple conditions made several older participants emotionally distressed and fearful of being a burden to their children. A few participants who had lost their spouse found it highly challenging to manage daily life and their MLTCs.

*"I live with my younger son. I have three children. I came alone that day (to FHC). Then, the ECG was taken, and two or three tablets were given. I was extremely uncomfortable for the last two days. I said to my son yesterday to check my blood sugar. . . he was busy and could not do it. I feel like I am a burden to them. Earlier, my son came twice and bought the medicines from health centre after showing the doctor the treatment book." (P13, 82-year-old female with four chronic conditions)*

*"I would not be tense if my husband were there, I feel like I have no one to go to when I am having problems or health issues. My son always says, Mom, do not say anything, and do not say anything. To whom will I tell my problems? What will I do?" (P26, a 56-year-old female with three chronic conditions)*

Younger participants found it difficult to accept that they had several conditions which added stress and concern to their lives. Younger participants were also worried about their work and family and expressed difficulties managing the multiple additional demands placed on them due to MLTCs.

*"I feel unfortunate because I have many conditions. I would not have felt this bad if it were just one condition. When I think about how all these conditions have affected me, I feel sad and tense at the same time. The atmosphere at home is also filled with unease as we worry about the potential consequences if any wounds develop or the conditions worsen. This fear is constantly present. My primary concern is diabetes, which poses the most significant challenge. It seems my blood sugar levels can be low today, but the next day they might be high." (P28, a 45-year-old female with four chronic conditions)*

Living with multimorbidity also affected the participants' emotional health, as they worried about the expenses associated with treatment for multiple conditions.

*"My major worry is that I do not have money with me. I feel sad to depend on my children for everything, for medicines, for going to hospital, and it is not like I must go once. Even, if I want to have tea, I should have money with me, or else how will I be tension-free?" (P3, a 64-year-old male with two chronic conditions)*

## Challenges to self-management

In this study, the self-management support is limited to verbal self-management information provided by HCPs during health care consultations as reported by the participants.

Most of the older participants reported following dietary practices that focused on reducing sugar, salt, and rice. However, younger participants found it difficult to make necessary nutritional changes due to the limited availability of food choices at work. Younger participants also reported that they may follow dietary restrictions for a shorter duration and stop them once their blood glucose or cholesterol levels are normal without advice from HCPs.

*I had two primary health issues, namely, cholesterol and alcoholic liver cirrhosis. To address these concerns, I made significant lifestyle changes, including completely cutting off sugar, including tea, from my diet for two months. Additionally, I quit smoking cigarettes three years ago. After experiencing an issue with ECG variation, I abstained from alcohol for approximately two months. Although I have not completely stopped alcohol from my life, I now only drink on social occasions, such as for the sake of company (friends). (P31, a 41-year-old male with four chronic conditions)*

Older participants with MLTCs faced difficulty doing recommended physical activities such as walking and reported tiredness and pain while walking. However, older participants without additional pain symptoms or breathlessness reported engaging in recommended walking by HCPs.

*I cannot go for any walks; both my legs have severe pain. I have had this pain for a long time, around 20 years. (P7, a 65-year-old female with three chronic conditions)*

*Some days I will walk in the front area of my house, whereas sometimes, I will be bedridden, then I might need help to take my medicine, my sister will come and help me. (P10, a 73-year-old female with three chronic conditions)*

Self-management was difficult for many patients due to their MLTCs. Many participants felt very restricted in their diet. Financial constraints also made it difficult to follow dietary recommendations for many patients, and they often felt unable to control their conditions.

*I do not know why it is high (blood glucose). I do not eat much rice, and I will not drink tea with sugar. I do not know; sometimes, I feel like this medicine is not effective. (P15, a 65-year-old female with five chronic conditions).*

## Care-coordination maze

The second theme captures how the participants deal with the requirements placed on them by the healthcare system to manage their multiple illnesses. Three subthemes were identified regarding participants' efforts in coordination of healthcare; fragmentation and poor continuity of care, medication management; an uphill battle and primary care falling short.

### Fragmentation and poor continuity of care

Most participants described difficulties in travelling to multiple health care facilities to seek care for multiple conditions. Depending on the nature and severity of their several long-term conditions, they were often treated by different HCPs in different settings, such as a tertiary care medical college, a private/public hospital, and FHCs. While most participants shared their challenges in traveling to medical colleges or other hospitals for consultations and routine medications, several participants in rural FHCs emphasized that these journeys and appointments could consume an entire day. Participants highlighted that often they would have to take multiple buses to reach medical colleges or hospitals making the journeys time consuming as well as expensive.

> *"The transportation fare is the problem. From here (participant's hometown) I will have to go to K (place name), then to T (place name) and finally to medical college and I have to travel in three buses. And also, it is a one entire day affair." (P2, a 71-year-old male with four chronic conditions)*

Many participants recollected their experiences of needing multiple visits to HCPs in several healthcare facilities and that information on their MLTCs or several medicines was not readily available. They elaborated on their struggle in visiting several doctors for their diagnosis, subsequent treatment during the acute phase, and continuing treatment at different centres.

> *"Initially, I experienced involuntary urination, which led me to seek medical help. I visited a private doctor in X private hospital, who referred me to another private hospital. I received medication there for a year, I do not think it was specific for kidney disease; they did not tell me for what. Next, I was advised to see a nephrologist at the medical college I was prescribed appropriate medication. But I will not get those medicines (kidney issues) at FHC, but I continue to take medication for diabetes from FHC." (P3, a 64-year-old male with two chronic conditions)*

Participants complained regarding the need for multiple attendances at different facilities with several specialists in public or private settings for their overall care. While most participants deemed specialists as knowledgeable and held high esteem for them, they found coordinating care between specialists and FHC difficult. They highlighted the limited accessibility to comprehensive healthcare services, which resulted in a burdensome and time-consuming process of seeking care.

> *"What else can I do? For pressure and sugar, I take medicines from here (FHC). Rest of the medicines I will not get here, so I must go to X (a tertiary private hospital) in Y (place) and see the doctor (stroke). Then only will I get to know if there are any changes. I have pain in both of my legs and my knees. For this, I must see an ortho at J private hospital (place A). If the doctor is not there, I must go to K private hospital in place B. . . Even if I do not want to go, we do not have any ortho doctor here (FHC), there is no point in explaining this to her (doctor at FHC)? I feel that as the ortho doctor is actually looking into my pain and joint issues and then prescribing, I will have to take the medicine they give (J, private hospital)." (P 23, a 46-year-old female with six chronic conditions)*

A few participants also raised a critical issue in accessing FHCs, particularly during the evening and nighttime, as FHCs are closed, leading to a lack of continuity of care. For example,

one individual highlighted that their health conditions tend to deteriorate during these hours, which can be a critical and concerning time for medical emergencies.

*Mostly, the conditions get worse in the evening or at night. Then it (FHC) will be closed, and we will have to go to a private (hospital) or Medical College and start explaining what happened. There is no one at home to help us also. (P2, a 71-year-old male with four chronic conditions)*

In contrast, most participants seeking care at private facilities or visiting the same doctors on multiple visits perceived it easy to communicate their issues effectively with the doctors. They acknowledged that effective HCP-to-patient handover communication ensured continuity of care for their long-term conditions.

*I consulted a private doctor, X. He said I am on the borderline of having blood pressure, and for now, it is not necessary to have medicines. Then later, I had a blood pressure of more than 140, and the same doctor said it is better to take medicines. After this, I had sugar, which started ten years ago. This doctor himself also identified sugar. During that time, I felt so tired, and I started becoming slimmer, so he said to do a blood test and found that I was also having sugar. So, the doctor told me to start medicines and explained how I should manage them. (P9, a 62-year-old male with three chronic conditions)*

## Medication management; an uphill battle

Several participants highlighted the substantial efforts required to manage their multiple medications. Some medications for chronic conditions such as diabetes, hypertension, chronic obstructive pulmonary diseases, and cardiovascular diseases are available at the FHCs. Since participants could not get all their medicines from a single point of care, they were compelled to purchase medicines from private pharmacies or return to the tertiary care centre. This limited availability of medications led to increased travel, higher expenses, and the added burden of remembering to procure medicines from various pharmacies.

*Only a few medicines are available here (FHC). Medicines prescribed by the medical college are not here (FHC). That is very costly medicine. . .I always have to make sure that I know which one I get from FHC, private pharmacy, and medical college. Sometimes, the FHC will not have stock, then I will have to come again. (P14, a 62-year-old male with three chronic conditions)*

The ability of participants to effectively handle the task of remembering, monitoring, and consistently taking their medications was impacted by their confusion regarding their prescribed drugs, as they often struggled to identify the various medicines they were required to take correctly. Additionally, older participants reported experiencing memory-related challenges, necessitating the implementation of additional strategies and techniques to ensure they took their medications as scheduled.

*Are you taking medicines regularly? If so, how? Interviewee: I have set an alarm on my phone. Interviewer: Who has set it? Is it your daughter? Interviewee: [. . .] It will ring at 9 in the morning, 1:30 in the afternoon, and nine at night. So, it will be there in my mind when I turn it off (that I should take medicine). It is the rule that I should not simply turn it off. (P5, a 62-year-old male with three chronic conditions)*

Several older participants occasionally encountered challenges in managing multiple medications, expressing concerns that these medications affected their physical capabilities and overall functioning. Conversely, younger adults expressed worries about the prolonged duration of medication use, primarily due to potential side effects.

*"What if all these medicines affect my kidney? Some patients say that they developed osteoporosis. But if we stop the medication, my sugar levels will double. I have not shared these concerns with anyone. I will keep it in mind itself. If I talk to my husband regarding this, he will tell me to stop the medication and regulate my diet." (P1, a 44-year-old female with two chronic conditions)*

Older participants maintained a regular monthly schedule of follow up visits to the FHCs for consultations with doctors who prescribed medications and conducted necessary lab tests to monitor complications. In contrast, younger participants revealed that they did not adhere to regular follow-ups with HCPs at FHCs, citing reasons such as time constraints or the absence of noticeable symptoms.

*After leaving the hospital, I could not get enough time to go back and follow-up. (P31, a 41-year-old male with four chronic conditions)*

*Diabetes was first detected when I went to Medical College for surgery (cauterization of warts on the leg, five years ago). I checked my blood sugar as the doctor had said, it was slightly high. Then, the doctor told me to control my diet. After a few days, it was normal and the surgery was done. After that, I did not check my blood sugar for long. After a few years, I had hernia surgery. Again, the doctor told me to check my blood sugar, and found that it was high. (P30, a 45-year-old male with two chronic conditions)*

### Primary care falling short

This sub-theme examined the participants' motivations for seeking health care from FHCs and how the FHCs responded to their needs. Many participants in this study were diagnosed and treated for chronic conditions such as cardiovascular diseases, chronic kidney diseases, bone or joint complaints, and respiratory issues at higher facilities, either in a public or private setting. For participants with more severe MLTCs, the FHC played a facilitative role in monitoring and providing monthly medications after consultations with doctors at the FHC. However, participants generally reported that they considered their specialist doctors as their primary HCPs and did not feel that FHCs could manage their different conditions. According to the participants, their primary purpose for visiting the FHC was to obtain monthly free medications for diabetes, hypertension, and any other available medications for their other long-term conditions.

*"To treat all these (several conditions), we need to have doctors here (FHC). It is not proper if we have only one doctor. If we have other (specialists) doctors, we can tell them." (P2, a 71-year-old male with four chronic conditions)*

Most participants expressed positive experiences with HCPs at FHCs, noting their cooperative and friendly nature. Participants felt that they were informed about the significance of effectively managing medication, diet, and physical activity to control their diabetes and hypertension by HCPs at FHC. Nevertheless, some participants were reluctant to discuss

problems related to stress or sleeplessness with their doctors, apprehensive that it might lead to additional medications being prescribed.

*"I never told the doctor about my issues with memory. Many people told me that it occurs due to low sugar levels. I am already taking many tablets. If I tell the doctor, he will give me more tablets. So, I will have to eat that too. That is why I never told the doctor." (P14, a 62-year-old male with three chronic conditions)*

Participants expressed difficulties in communicating their concerns with HCPs. They felt that the doctors did not have enough time to listen to their concerns owing to the long queues.

*"Then there is a long waiting time, so I could not say everything to the doctor." (P10, a 73-year-old female with three chronic conditions)*

*Usually, they do not explain anything. Last time, the tablets I received were old, and started to break down when I opened them. I wanted to ask them about that. (P1, a 44-year-old female with two chronic conditions)*

## Discussion

This study shows that patients with MLTCs interviewed from Kerala experienced considerable impact on their daily activities. This was further exacerbated with care being provided by multiple providers, taking multiple medications, the demands of self-management, increased travel, costs, of care, and the cognitive workload required for care coordination. A recent secondary phenomenological analysis from 10 LMICs showed similar findings to our study in that burden of multimorbidity treatment is compounded by fragmented healthcare, leading to duplicated care, higher costs, and lack of clarity in the treatment. Further, both patients and carers bear significant responsibilities in managing these conditions [34].

### Conditions related work-load

While there is limited information about living with MLTCs in Kerala and India [35], previous studies from India have noted mental distress as both a contributor to and a consequence of long-term conditions like diabetes [36, 37]. Our study further highlighted experiences of emotional distress among both older and younger patients living with MLTCs from Kerala. Older individuals expressed emotional distress due to their reliance on their children and financial instability. In contrast, younger adults struggled with accepting their multiple conditions and had concerns about the long-term use of multiple medications, which contributed to their emotional distress. This contrast might be attributed to older participants accepting their illness and adapting to their health management routines over time.

Involving family members in care remains a valuable part of healthcare in India [38]. However, the presence of MLTCs could amplify emotional difficulties associated with expectations and the burden of care for people living with MLTCs. These physical and emotional challenges can significantly impact patients' quality of life and increase the risk of complications and poor prognosis. Studies from high-income countries (HICs) have also reported physical and psychological difficulties, especially among older patients with MLTCs [39–41]. There is convincing evidence indicating that addressing specific risk factors, such as mental health issues in individuals with MLTCs, leads to improved mental health [42]. This emphasizes the importance of regular screening for mental health issues in patients with MLTCs and considers the implementation of both non-pharmacological and pharmacological management of these

conditions in primary care settings. The reforms under the "Aardram mission" established additional services such as depression screening clinics (ASWAAS, meaning "assurance") at the FHC level [18]. The doctors and nurses at FHCs are trained to screen for mental health issues and manage them using psychosocial counselling and clinical guidelines [19]. The findings in this study show ways in which emotional issues contribute to illness burden and underscore the need to recognize that individuals living with MLTCs maybe considered high risk or more prone to developing difficulties in coping with their conditions and hence the need to offer better mental health support at FHCs.

## Health system related work-load

While we did not set out to test any models of treatment burden, themes and sub-themes relating to how the health system adds to the workload of individuals with MLTCs in this study are consistent with the literature on treatment burden. Treatment burden pertains to the healthcare workload that individuals with chronic illnesses endure and how it impacts their overall functioning and well-being [43]. Despite extensive research previously conducted on the treatment burden in HICs for various chronic conditions and MLTCs, there is a noticeable absence of information regarding patient perceptions of managing MLTCs within the healthcare systems in India [44–46]. Drawing from a study of heart failure patients in the UK (United Kingdom) by Gallacher et al., it is important to acknowledge that disparities in healthcare systems can lead to differences in the treatment burden experienced by patients [47]. Our study sheds light on how patients with MLTCs from Kerala are compelled to assume a more significant role in coordinating their care across various healthcare facilities and providers, managing their multiple medications and thus imposing significant demands on their time and lives.

Previous research from India has highlighted the challenges patients face in finding affordable, reliable, and trustworthy healthcare HCPs when dealing with single chronic conditions [48, 49]. Our study builds upon these findings by revealing how participants living with MLTCs must independently navigate the management of multiple providers, treatment plans, and appointments. Our study demonstrated patients facing the added burden of travel, memory, communication, and the quest for affordable medicines to address their MLTCs. A prior study from Malawi, exploring the experiences of patients with multimorbidity, introduced another facet of treatment burden: the "burden of lack of treatment." This dimension encapsulates the emotional burden of knowing what is missing in one's treatment and that these treatments are not available due to a lack of economic resources or services in nearby health centres [50]. Although we did not specifically detect the issue of worry about treatment missing, our study highlights the cognitive burden of remembering, seeking and buying multiple medications from several centres due to the unavailability of medicines in primary care and coordinating care. These insights highlight the need for healthcare systems to address these complexities to better support patients with MLTCs.

Given the nature of MLTCs, receiving care from several HCPs is often required. However, to address the challenges reported by patients, a change in the organization of care delivery, typically through case management or enhanced multidisciplinary teamwork, may be beneficial. [42, 51] The healthcare system in Kerala has implemented significant changes to strengthen primary care, including the recent addition of mid-level health providers (MLHPs) to the primary care workforce through the national "Ayushman Bharat" program [52]. The focus of MLHPs is to improve the prevention and management of NCDs. Therefore, interventions in case management involving MLHPs and the primary health care team at the FHCs can contribute to better coordination of services. For instance, the government-sponsored e-Sanjeevani OPD telemedicine service offers free teleconsultations. However, many patients

may be unaware of this facility or lack the technical skills to utilize it effectively. MLHPs can serve as a crucial link between patients and specialists, helping to organize teleconsultations for patients at the sub-centre level, which has been recently included in MLHPs' job responsibilities (GO-Rt-No 131-2023-H&FWD Sub Centre Functions, dated 19 January 2023 by the Government of Kerala). MLHPs can play a vital role in HCP-to-patient handover communication and explaining self-management strategies to patients with the aid of patient-held records [53]. This approach is particularly relevant since several studies have established that doctors in outpatient settings in the Kerala public health system often have limited time for patient consultations [54, 55].

## Self-management related workload

Self-management workloads experienced in our study include adhering to dietary recommendations, adhering to medication regimens and follow-up, and engaging in physical activity. While patients with MLTCs reported receiving information on self-management, there appears to be a difficulty in tailoring this information for self-management practices due to the illness burden experienced. Our study findings are consistent with previous research that communicative and critical health literacy play substantive role than functional health literacy in fostering self-management behaviours in chronic conditions [56, 57]. Health literacy is viewed as comprising three progressive skills: functional (reading and writing), communicative (cognitive and literacy skills for applying health information), and critical health literacy (the ability to analyze and use data for health decisions) [58, 59] However, it is also crucial to recognize that the healthcare systems [60]can significantly mitigate the potential mismatch between low health literacy among patients and the system by providing personalized care, easily understandable communication, and tailored support.

Although individuals faced challenges in self-management, the perceived reasons for these difficulties varied among participants. Both younger and older individuals encountered issues with adhering to physical activity, medications, and follow-up healthcare visits. Therefore, beyond health system support, those with MLTCs may benefit from a network of social relationships to manage their healthcare and lives. [56, 61] Improved access to a social support network has been demonstrated to enhance mental health and treatment adherence [56, 57] A recent study from Kerala highlighted the role of neighbourhood clinics in improving healthcare access and building social capital for older men and women in the areas they served. Operating under the name *Vayomitram*, these neighbourhood clinics were situated where many individuals aged 65 and above resided, offering preventive and curative primary care. These clinics operated out of government day-care centres (Anganwadi's) and other schools in the area. An essential function of these neighbourhood clinics was addressing the social and psychological needs of the older population by providing a space for them to gather and build social networks. This, in turn, contributed to improved well-being and access to other resources long after the mobile clinic had left the neighbourhood [62].

## Implications

Our findings, lead to the potential implications below for India and similar health systems that are not currently organized/prepared to manage MLTCs. Given the increasing burden of non-communicable MLTCs in Kerala and, more broadly, in India, this study amplifies the long-standing need for good quality, reliable, and easily accessible routine primary care services. These services should be made available even in the evening hours to cater to the needs of the working population. To protect people from financial burden due to managing their MLTCs, provisions of free or subsidized medicines should be available at a single point of care. Overall,

considering the additional treatment burden for patients with MLTCs, their treatment needs must be prioritized with better patient-HCP interaction. Lastly, improvements in social networks and social welfare coverage, such as access to physiotherapy, home support for healthcare monitoring etc., are necessary to improve treatment adherence and quality of life. Further research on building capacities of individuals with MLTCs within their social networks, with emphasis on primary care, is needed for developing patient-centred interventions for managing long-term conditions.

## Strengths and limitations

We conducted the study across different health centres in three districts in Kerala, leading to diverse participant perceptions from more expansive geographical locations, increasing the potential for transferability of findings to other communities in Kerala and similar health systems. The study represents one of the few studies that has examined the experiences of adults living with MLTCs in India. An interdisciplinary triangulation of researchers provided a broader disciplinary perspective and contributed to the rigor of the research process. A limitation of this study is that interviews were conducted in the public health centres and hence may have excluded those who do not access FHC. In two instances, interviews ended prematurely due to patients wanting to return to their queues for consultation.

## Conclusion

Our study adds to the growing body of research on the experiences of individuals with MLTCs in India and LMICs more broadly. Notably, our findings emphasize an additional workload beyond the burden of living with MLTCs as patients grapple with challenges related to accessing affordable medications and services and coordinating their care within the health system. These factors should be considered when designing interventions to enhance the care provided to people living with MLTCs. Our research demonstrates that both younger and older individuals with MLTCs experience the burden of illness and treatment. However, our findings reveal distinct reasons for this burden, suggesting the need for tailored and patient-centred approaches to manage MLTCs. By recognizing and addressing these distinct needs, healthcare interventions can better address the diverse challenges faced by individuals living with MLTCs, ultimately fostering improved health outcomes and quality of life.

## Supporting information

**S1 File.**
(DOCX)

## Author Contributions

**Conceptualization:** Jissa Vinoda Thulaseedharan, Mathew Joseph Valamparampil, Sivadasanpillai Harikrishnan, Sheila Greenfield, Paramjit Gill, Justine Davies, Semira Manaseki-Holland, Panniyammakal Jeemon.

**Data curation:** Linju Joseph, Athira Krishnan, Thoniparambil Ravindranathanpillai Lekha, Neethu Sasidharan.

**Formal analysis:** Linju Joseph, Athira Krishnan, Thoniparambil Ravindranathanpillai Lekha, Neethu Sasidharan, Sheila Greenfield, Panniyammakal Jeemon.

**Funding acquisition:** Jissa Vinoda Thulaseedharan, Mathew Joseph Valamparampil, Sivadasanpillai Harikrishnan, Paramjit Gill, Justine Davies, Semira Manaseki-Holland.

**Investigation:** Linju Joseph.

**Methodology:** Linju Joseph, Thoniparambil Ravindranathanpillai Lekha, Sheila Greenfield.

**Project administration:** Linju Joseph, Semira Manaseki-Holland, Panniyammakal Jeemon.

**Supervision:** Jissa Vinoda Thulaseedharan, Sheila Greenfield, Paramjit Gill, Justine Davies, Semira Manaseki-Holland, Panniyammakal Jeemon.

**Writing – original draft:** Linju Joseph.

**Writing – review & editing:** Linju Joseph, Athira Krishnan, Thoniparambil Ravindranathanpillai Lekha, Jissa Vinoda Thulaseedharan, Mathew Joseph Valamparampil, Sivadasanpillai Harikrishnan, Sheila Greenfield, Paramjit Gill, Justine Davies, Semira Manaseki-Holland, Panniyammakal Jeemon.

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
