## [Decision Letter · Decision Letter 0]

9 Apr 2024

PONE-D-24-05334Experiences and challenges of people living with multiple long-term conditions in managing their care in primary care settings in Kerala, India: A qualitative studyPLOS ONE

Dear Dr. Jeemon,

Thank you for submitting your manuscript to PLOS ONE. After careful consideration, we feel that it has merit but does not fully meet PLOS ONE’s publication criteria as it currently stands. Therefore, we invite you to submit a revised version of the manuscript that addresses the points raised during the review process.

We look forward to receiving your revised manuscript.

Kind regards,

Shekhar Chauhan

Academic Editor

PLOS ONE

Journal Requirements:

"A research grant from the Medical Research Council UK funded this work (MC_PC_MR/T037822/1). "

4. We note that your Data Availability Statement is currently as follows: [All relevant data (excerpts from the interview transcripts) are within the manuscript and its supporting information files.]

Additional Editor Comments:

Dear Authors,

This is magnificent work by you and your team. However, I have some suggestions that you should consider incorporating while submitting the revised version of the manuscript.

1. We recommend that authors use the COREQ checklist, or other relevant checklists listed by the Equator Network, such as the SRQR, to ensure complete reporting (http://journals.plos.org/plosone/s/submission-guidelines#loc-qualitative-research). In general, we would expect qualitative studies to include the following: 1) defined objectives or research questions; 2) description of the sampling strategy, including rationale for the recruitment method, participant inclusion/exclusion criteria and the number of participants recruited; 3) detailed reporting of the data collection procedures; 4) data analysis procedures described in sufficient detail to enable replication; 5) a discussion of potential sources of bias; and 6) a discussion of limitations.

2. Use the term 'multimorbidity' along side MLTC in the abstract- same way as you are using in the opening sentence of the manuscript (background).

3. At places, you have cited NFHS-4 related findings (references). Just a suggestion, if you could find and cite the NFHS-5 as it is recent one.

4. Conclusion section could be strengthened more.

Reviewers' comments:

Reviewer's Responses to Questions

**Comments to the Author**

1. Is the manuscript technically sound, and do the data support the conclusions?

Reviewer #1: Yes

Reviewer #2: Yes

2. Has the statistical analysis been performed appropriately and rigorously? 

Reviewer #1: N/A

Reviewer #2: N/A

3. Have the authors made all data underlying the findings in their manuscript fully available?

Reviewer #1: Yes

Reviewer #2: Yes

4. Is the manuscript presented in an intelligible fashion and written in standard English?

Reviewer #1: Yes

Reviewer #2: Yes

5. Review Comments to the Author

Reviewer #1: Well written and an important subject. Themes are also well developed. Just suggestion to include in the table details of the conditions that persons that were interviewed had.

Also consider this paper on multi morbidity experience in LMICs that has similar themes to what was found in the study.

Tran PB, Ali A, Ayesha R on behalf of the COSMOS collaboration, et al An interpretative phenomenological analysis of the lived experience of people with multimorbidity in low- and middle-income countries BMJ Global Health 2024;9:e013606.

Reviewer #2: Thank you for asking me to review this interesting manuscript on the Experiences and challenges of people living with multiple long-term conditions in managing their care in primary care settings in Kerala.

Overall, this is a good study and well written manuscript. However, there are some areas that would benefit from further revision or additional details.

Abstract: please add to the abstract that this is a qualitative descriptive study and report that you used a framework approach for analysis. Please report the themes more clearly. I would suggest removing reference to a ‘comprehensive understanding’ of this problem and suggest replacing with ‘in-depth’ understanding.

I would prefer to see reference to ‘the lived experience’ removed from the abstract and replaced with ‘Specifically, it explores how people living with MLTCs describe their illness, their interactions with healthcare services, and their challenges in primary care settings in Kerala, India.’

Introduction: Well written and articulates the problem clearly.

Line 101 ‘Primary care settings in LMICs often fail to consider the patient's perspective’ what is this statement based on?

Method: Please report within this section that you have reported this study in accordance with the SRQR checklist.

Methodology: Although the authors describe this as a ‘descriptive qualitative study’ I would like to see more details about the philosophical/theoretical positioning of the study, what is the study’s ontological and epistemological position? Is the study inductive or deductive? Also, some defence is required for why the study was positioned as QD and consideration of it benefits and limitations. The reference used as support for the use of a qualitative descriptive approach is to a paper by Neergaard at al. However, I would suggest reading and citing the original description of QD by Sandelowski 2002.

Sampling: ‘Recruitment continued until theoretical saturation, indicating that additional participants were unlikely to provide significantly new information’ - this is a very problematic sentence. Theoretical saturation is associated with grounded theory methodology and accompanies theoretical sampling not purposive and has an approach that combines data analysis with sampling, so the researchers know when to stop recruiting. It is not clear if this is what you did. Please either qualify this statement, or add further detail on how you were confident that your sample was appropriate to answer the research question.

Data collection: I would like to see the inclusion of a specific reflexivity statement where each researcher’s background is described. I would like to know why four people were used as interviewers, if these were conducted individually or in teams and if dividing the data collection up like this had any positive or negative consequences. Thank you for including the interview schedule. However, please say how this was developed. Did you have any CEI/PPI to develop the methods for this study?

Data analysis: In the QD reference cited in the methods by Neergaard at al. a QD analysis method is provided by Miles et al. Why did you therefore choose a Framework approach? You need to say how the approach by Gale et al is commensurate with a QD approach. Please reference the stages to Gale et al on line 141.

Within the main text please provide further explanation of stages 3 & 4 with greater explanation of ‘open coding’ (e.g. was this inductive or deductive) and the process of ‘concordance’.

I would also like to see a specific heading of ‘quality’ and some discussion of steps used, in accordance with QD and framework analysis, that ensure the study is rigorous.

Findings: the themes feel unbalanced in that one has two sub-themes, one has four and one has none. The theme titles also feel underdeveloped, I think these could be refined further to convey a better understanding of the problem.

6. PLOS authors have the option to publish the peer review history of their article (what does this mean?). If published, this will include your full peer review and any attached files.

Reviewer #1: No

Reviewer #2: No

---

## [Author Response · Author response to Decision Letter 0]

2 May 2024

Authors’ response 

We would like to thank the editor and reviewers for a careful and thorough reading of our manuscript titled “Experiences and challenges of people living with multiple long-term conditions in managing their care in primary care settings in Kerala, India: A qualitative study.” We appreciate the thoughtful comments and constructive suggestions, which have helped to improve the quality of the manuscript. 

Our point-by-point responses for reviewer comments are presented in italics. The page numbers and line numbers referred in this document are based on the tracked change version of the manuscript. 

Sincerely,

On behalf of authors

Linju Joseph

Editor’s comments

Dear Authors,

This is magnificent work by you and your team. However, I have some suggestions that you should consider incorporating while submitting the revised version of the manuscript.

Thank you for your encouraging feedback.

1. We recommend that authors use the COREQ checklist, or other relevant checklists listed by the Equator Network, such as the SRQR, to ensure complete reporting (http://journals.plos.org/plosone/s/submission-guidelines#loc-qualitative-research). In general, we would expect qualitative studies to include the following: 1) defined objectives or research questions; 2) description of the sampling strategy, including rationale for the recruitment method, participant inclusion/exclusion criteria and the number of participants recruited; 3) detailed reporting of the data collection procedures; 4) data analysis procedures described in sufficient detail to enable replication; 5) a discussion of potential sources of bias; and 6) a discussion of limitations.

 Response

Thank you for your suggestion. We had already uploaded the SRQR (Standards for Reporting Qualitative Research) checklist and now we have incorporated the following in the main manuscript.

“We have drawn on the enhancing transparency in reporting the qualitative study using SRQR (Standards for Reporting Qualitative Research) checklist.” (page number: line numbers)

2. Use the term 'multimorbidity' alongside MLTC in the abstract- same way as you are using in the opening sentence of the manuscript (background).

Response

We have incorporated this suggestion and now the abstract reads as follows “Multimorbidity or multiple long-term conditions (MLTCs), the coexistence of two or more chronic conditions within an individual, presents a growing concern for healthcare systems and individuals' well-being.”

3. At places, you have cited NFHS-4 related findings (references). Just a suggestion, if you could find and cite the NFHS-5 as it is recent one.

Response

We have incorporated this suggestion and the introduction reads “A cross-sectional study the National Family Health Survey (NFHS)-4 reported a prevalence of 7.2% for MLTCs among adults aged 15-49 years in India and NFHS -5 showed that the prevalence of multimorbidity increased with advancing age. [12, 13]” page number 4, line numbers 86-89

4. Conclusion section could be strengthened more.

Response

Thank you for your suggestion.

“Our study adds to the growing body of research on the experiences of individuals with MLTCs in India and LMICs more broadly. Notably, our findings emphasize an additional workload beyond the burden of living with MLTCs as patients grapple with challenges related to accessing affordable medications and services and coordinating their care within the health system. These factors should be considered when designing interventions to enhance the care provided to people living with MLTCs. Our research demonstrates that both younger and older individuals with MLTCs experience the burden of illness and treatment. However, our findings reveal distinct reasons for this burden, suggesting the need for tailored and patient-centred approaches to manage MLTCs. By recognizing and addressing these distinct needs, healthcare interventions can better address the diverse challenges faced by individuals living with MLTCs, ultimately fostering improved health outcomes and quality of life.” Page number28, line numbers 616-626. 

Reviewers' comments:

1. Reviewer #1: Well written and an important subject. Themes are also well developed. Response

Thank you for the encouraging feedback on our manuscript.

2. Just suggestion to include in the table details of the conditions that persons that were interviewed had.

Response

We have included the details of conditions and other person-specific demographic details in the online supplement as Table S1.

3. Also consider this paper on multi morbidity experience in LMICs that has similar themes to what was found in the study. Tran PB, Ali A, Ayesha R on behalf of the COSMOS collaboration, et al An interpretative phenomenological analysis of the lived experience of people with multimorbidity in low- and middle-income countries BMJ Global Health 2024;9:e013606.

Response

Thank you for this suggestion. We have now incorporated the following in the main manuscript. 

“A recent secondary phenomenological analysis from 10 LMICs showed similar findings to our study in that burden of multimorbidity treatment is compounded by fragmented healthcare, leading to duplicated care, higher costs, and lack of clarity in the treatment.” (page number 22, line numbers 483-487)

Reviewer #2: Thank you for asking me to review this interesting manuscript on the Experiences and challenges of people living with multiple long-term conditions in managing their care in primary care settings in Kerala.

Overall, this is a good study and well written manuscript. However, there are some areas that would benefit from further revision or additional details.

Response

Thank you for the encouraging feedback on our manuscript. 

1. Abstract: please add to the abstract that this is a qualitative descriptive study and report that you used a framework approach for analysis. Please report the themes more clearly. I would suggest removing reference to a ‘comprehensive understanding’ of this problem and suggest replacing with ‘in-depth’ understanding.

Response

We have incorporated this suggestion and now the abstract reads as follows “We designed a qualitative descriptive study and conducted in-depth, semi-structured interviews with 31 people (16 males and 15 females) from family health centres (FHCs) in Kerala; Interview data were recorded, transcribed, and thematic analysis using the Framework Method was undertaken.” (page number 2, line numbers 37-40)

We have re-organised and re-named themes and subthemes to convey better understanding of the problem. We have incorporated the following in the manuscript “Two main themes and three sub-themes each were identified; (1) Illness impacts on life (a)physical issues (b) psychological difficulties (c) challenges of self-management and (2) Care-coordination maze (a)fragmentation and poor continuity of care (b) medication management; an uphill battle and (c) primary care falling short.” (page number 2, line numbers 42-45)

“This study offers an in-depth view of the experiences of individuals living with MLTCs from Kerala, India.” (page number 3, line number 59)

2. I would prefer to see reference to ‘the lived experience’ removed from the abstract and replaced with ‘Specifically, it explores how people living with MLTCs describe their illness, their interactions with healthcare services, and their challenges in primary care settings in Kerala, India.’

Response

We have incorporated this suggestion and now the abstract reads as follows “We explore how people living with MLTCs describe their illness, their engagements with healthcare services, and challenges they face within primary care settings in Kerala, India.” (page number 2, line numbers 32-34)

3. Introduction: Well written and articulates the problem clearly.

Line 101 ‘Primary care settings in LMICs often fail to consider the patient's perspective’ what is this statement based on? 

Response

We have amended the sentence as follows. 

Policies on primary care settings or NCDs in LMICs often fail to consider the patient's perspective, which is crucial for understanding their experiences of MLTCs and addressing broader social and economic factors that impact healthcare delivery, patient outcomes, and prognosis. (page number 5, line numbers 106)

4. Method: Please report within this section that you have reported this study in accordance with the SRQR checklist.

Response

We have incorporated this suggestion. We have reported the qualitative study using SRQR (Standards for Reporting Qualitative Research) checklist. (page number 6, line numbers 134-136)

5. Methodology: Although the authors describe this as a ‘descriptive qualitative study’ I would like to see more details about the philosophical/theoretical positioning of the study, what is the study’s ontological and epistemological position? Is the study inductive or deductive? Also, some defence is required for why the study was positioned as QD and consideration of it benefits and limitations. The reference used as support for the use of a qualitative descriptive approach is to a paper by Neergaard at al. However, I would suggest reading and citing the original description of QD by Sandelowski 2002.

Response

Thank you for your suggestion and find the below justification for qualitative descriptive approach for our study which we have added in the manuscript. (page number 6, line numbers 119-134)

 “Given the limited existing research on the subject, a qualitative descriptive approach was deemed appropriate. This methodology allows researchers to remain closely connected to the raw data, facilitating the emergence of insights into the phenomenon under investigation.[23] By adopting this approach, the study sought to provide an in-depth summary of the phenomenon through the analysis and interpretation of the meanings’ individuals attribute to events and supported by reference to verbatim quotations from participants.[24] The overarching aim of the project was to enhance healthcare practice by developing an intervention model for managing patients with MLTCs. 

Adopting a flexible design such as qualitative description enables data collection and analysis to be an iterative process by responding to participants' responses to questions and simultaneously adapting the analytical process as new insights emerge during the study.[25] Given the emphasis on providing a rich, descriptive account in qualitative descriptive design, thematic analysis using the Framework Method was chosen.[26,27] The Framework Method is employed for its suitability for use in multidisciplinary research teams and flexibility for both inductive and deductive analysis. It allows for a systematic exploration of different views or experiences of participants in relation to each topic, which can then be compared and contrasted effectively. We have reported the qualitative study using SRQR (Standards for Reporting Qualitative Research) checklist.”

6. Sampling: ‘Recruitment continued until theoretical saturation, indicating that additional participants were unlikely to provide significantly new information’ - this is a very problematic sentence. Theoretical saturation is associated with grounded theory methodology and accompanies theoretical sampling not purposive and has an approach that combines data analysis with sampling, so the researchers know when to stop recruiting. It is not clear if this is what you did. Please either qualify this statement, or add further detail on how you were confident that your sample was appropriate to answer the research question.

Response

Thank you for seeking clarification. Although grounded theory methodology was not explicitly employed in our study, we embraced the concept of theoretical or meaning saturation in a broader sense, as delineated by Hennink et al. Meaning saturation denotes the point at which issues are fully understood, and no further dimensions, nuances, or insights can be found. Our data collection and analysis were iterative, and we continued recruitment until we observed redundancy in the information provided by participants. This redundancy suggested that additional data collection was unlikely to yield substantially new insights relevant to our research question. 

In our revised statement, we emphasize that the research team ensured not only code saturation but also a deep and rich understanding of the themes derived during analysis (meaning saturation). (page number 9, line numbers 177-178)

7. Data collection: I would like to see the inclusion of a specific reflexivity statement where each researcher’s background is described. I would like to know why four people were used as interviewers, if these were conducted individually or in teams and if dividing the data collection up like this had any positive or negative consequences. Thank you for including the interview schedule. However, please say how this was developed. Did you have any CEI/PPI to develop the methods for this study?

Response

As reported in the manuscript, this qualitative study was a part of a larger project with the aim of developing a patient centred model for managing patients with NCD multimorbidity; multiple interviewers were required for the timely completion of the project. To ensure reliability of the data collected through multiple interviewers, all researchers were trained on interview topic guide. Additionally, incorporating both experienced and novice qualitative researchers enriched the study by capturing diverse perspectives. [32,33] The iterative approach to data collection and analysis ensured thorough engagement with the data, enhancing the credibility and depth of insights gathered. We did not have any CEI/PPI to develop the methods.

The following has been incorporated in the manuscript. (page number 9, line numbers 181-199)

“Methodological rigour and the research team

We acknowledge that using Framework Method, the analysis and subsequent themes were influenced by the research team's subjective interpretations of the data. However, throughout the analytical process, researcher reflexivity and audited discussions occurred between authors throughout the data collection, analysis and write-up to ensure rigour in the quality of qualitative analysis conducted. Specifically, charting enabled discussion among the diverse research team and analyst triangulation was ensured, which promoted objectivity between the researcher's position and the analysis. Charting made it easy to identify relevant quotes to illustrate themes from a range of participants and served as an audit trail from raw data to final themes. 

 LJ was a female, early career researcher with specific interests in intervention development and patient experiences. LTR was a female sociologist, with research expertise and interests in healthcare. AK and NS were female research assistants with background in public health and nursing. PJ was a researcher with expertise in chronic disease epidemiology and health systems interventions. LJ, LTR, AK, NS and PJ are native of Kerala. SG is a professor of medical sociology with expertise in cross-cultural research from UK. The other team members (JVT, MJV, SH, PG, JD and SMH) had expertise in clinical and/ health systems research.”

“The topic guide was prepared by consulting the research team and reviewing the literature on multimorbidity.”

8. Data analysis: In the QD reference cited in the methods by Neergaard at al. a QD analysis method is provided by Miles et al. Why did you therefore choose a Framework approach? You need to say how the approach by Gale et al is commensurate with a QD approach. Please reference the stages to Gale et al on line 141.

Response

Please refer to response to question number 5.

9. Within the main text please provide further explanation of stages 3 & 4 with greater explanation of ‘open coding’ (e.g. was this inductive or deductive) and the process of ‘concordance’.

Response

Thanks for asking for explanation for the stages in analysis. Please find the below text which has been added in the manuscript. (page numbers 7-8, line numbers 167-174)

“Open or unrestricted coding facilitated excerpts from the interview data to be labelled inductively. Additionally, deductive codes from the topic guide were also included i

---

## [Editor Report · Decision Letter 1]

15 May 2024

PONE-D-24-05334R1Experiences and challenges of people living with multiple long-term conditions in managing their care in primary care settings in Kerala, India: A qualitative studyPLOS ONE

Dear Dr. Jeemon,

Thank you for submitting your manuscript to PLOS ONE. After careful consideration, we feel that it has merit but does not fully meet PLOS ONE’s publication criteria as it currently stands. Therefore, we invite you to submit a revised version of the manuscript that addresses the points raised during the review process.

**I am happy to see that you have carried out the revisions as required. I have further concerns about the manuscript and feel that you should consider them. **==============================

We look forward to receiving your revised manuscript.

Kind regards,

Shekhar Chauhan

Academic Editor

PLOS ONE

Journal Requirements:

**Additional Editor Comments:**

Dear Authors,

Thank you for carrying out the revision. The manuscript is now well structured. However, I have some points to raise before I can make a decision on your submission.

1. In table 1, there is no need to provide the initials of authors. You can consider moving the detailed table as supplementary file and can provide only the information without authors initials.

2. I feel that you should also provide the title to figure 1 within the figure itself. As of now, the title is there as a text in the manuscript. Delete that title text from the manuscript and include that within the figure.

---

## [Author Response · Author response to Decision Letter 1]

18 May 2024

Authors’ response 

We would like to thank the editor for a careful and thorough reading of our manuscript titled “Experiences and challenges of people living with multiple long-term conditions in managing their care in primary care settings in Kerala, India: A qualitative study.” Our point-by-point responses for comments are presented in italics. The page numbers and line numbers referred in this document are based on the tracked change version of the manuscript. 

Sincerely,

On behalf of authors

Linju Joseph

Additional Editor Comments:

Dear Authors,

Thank you for carrying out the revision. The manuscript is now well structured. However, I have some points to raise before I can make a decision on your submission.

1. In table 1, there is no need to provide the initials of authors. You can consider moving the detailed table as supplementary file and can provide only the information without authors initials.

Response

Thank you for your suggestion. We have now moved Table 1 to online supplement and have removed author’s initials based on your suggestion. Page number 8, line numbers 175-176; Online supplement page number 1.

2. I feel that you should also provide the title to figure 1 within the figure itself. As of now, the title is there as a text in the manuscript. Delete that title text from the manuscript and include that within the figure.

Response

Thank you for your suggestion. We have now deleted the figure title from the manuscript and have incorporated it in the figure itself.

---

## [Editor Report · Decision Letter 2]

30 May 2024

Experiences and challenges of people living with multiple long-term conditions in managing their care in primary care settings in Kerala, India: A qualitative study

PONE-D-24-05334R2

Dear Jeemon,

We’re pleased to inform you that your manuscript has been judged scientifically suitable for publication and will be formally accepted for publication once it meets all outstanding technical requirements.

Kind regards,

Shekhar Chauhan

Academic Editor

PLOS ONE
---

## [Editor Report · Acceptance letter]

3 Jun 2024

PONE-D-24-05334R2 

PLOS ONE

Dear Dr. Jeemon, 

I'm pleased to inform you that your manuscript has been deemed suitable for publication in PLOS ONE. Congratulations! Your manuscript is now being handed over to our production team.

Kind regards, 

on behalf of

Dr. Shekhar Chauhan 

Academic Editor

PLOS ONE